# Variables Impacting the Quality of Care Provided by Professional Caregivers for People with Mental Illness: A Systematic Review

**DOI:** 10.3390/healthcare10071225

**Published:** 2022-06-30

**Authors:** Lluna M. Bru-Luna, Manuel Martí-Vilar, César Merino-Soto, Guillermo Salinas-Escudero, Filiberto Toledano-Toledano

**Affiliations:** 1Departamento de Psicología Básica, Universitat de València, Avgda, Blasco Ibañez, 21, 46010 Valencia, Spain; llunamaria.bl@gmail.com (L.M.B.-L.); manuel.marti-vilar@uv.es (M.M.-V.); 2Instituto de Investigación de Psicología, Universidad de San Martín de Porres, Avenue Tomás Marsano 232, Lima 34, Peru; sikayax@yahoo.com.ar; 3Centro de Estudios Económicos y Sociales en Salud, Hospital Infantil de México Federico Gómez National Institute of Health, Márquez 162, Doctores, Cuauhtémoc, Mexico City 06720, Mexico; guillermosalinas@yahoo.com; 4Unidad de Investigación en Medicina Basada en Evidencias, Hospital Infantil de México Federico Gómez National Institute of Health, Márquez 162, Doctores, Cuauhtémoc, Mexico City 06720, Mexico; 5Unidad de Investigación Sociomédica, Instituto Nacional de Rehabilitación Luis Guillermo Ibarra Ibarra, Calzada México-Xochimilco 289, Arenal de Guadalupe, Tlalpan, Mexico City 14389, Mexico

**Keywords:** systematic review, mental illness, professional caregiver, psychological variables

## Abstract

People with mental illness may need the support of caregivers in certain areas of their lives, and there is an increasing need for quality care for people with mental health problems by qualified health professionals. Often, these professionals may develop so-called burnout syndrome, although some authors point out that positive emotions may also arise. In addition, several variables can act as both protectors and stressors. Therefore, the main aim of the current study is to identify variables related to the professional care of people with mental illness (i.e., protective or stressor variables) through a systematic review. The review was conducted according to the PRISMA guidelines with a final selection of 20 articles found in the Web of Science, PubMed, ScienceDirect and Dialnet databases between the months of October and November 2019, and updated in June 2022. The results show that job satisfaction is a strong predictor of the quality of care, and that congruence between personal and organizational values is a very important factor. Meanwhile, working in the same job for successive years, working in community mental health teams and experiencing burnout act as stressors and reduce the quality of care provided.

## 1. Introduction

With the arrival of the 21st century and changes to family structures, there is a growing need to invest resources to meet the demands of people in situations of dependency. According to the International Classification of Functioning, Disability and Health (ICF), dependency is defined as the result of a deficit in bodily function (i.e., physiological functions of the body systems, including psychological functions), bringing with it a limitation in activity. When environmental adaptation is unable to compensate for this limitation, there is a restriction in participation that causes the individual to depend on the help of others to carry out the activities of daily life [1]. There are several causes that can be the origin of a person’s dependence, the most common being aging, accidents or chronic illnesses, and they can give rise to different types of dependence, such as physical, mental or psychic, sensory or mixed dependence.

The American Psychiatric Association [2] defines mental illness as a significant alteration of emotional, cognitive and/or behavioral type. Basic psychological processes such as emotion, motivation, cognition, awareness, behavior, perception, learning and language are often affected. Such changes may affect personal care, the performance of domestic tasks, interpersonal relationships, professional functioning, and participation in leisure activities or community life. In addition, mental illness can have profound implications for subjective distress, psychological well-being, and a person’s quality of life [3]. All these factors can lead to a decompensation with the environment and cause situations of dependency in people with mental illness, especially in those with severe mental illness.

In the epidemiological domain, the Confederation of Mental Health in Spain [4] presented WHO data on the global prevalence and incidence of mental illness: 12.5% of all health problems are represented by mental disorders; in addition, this incidence increased by 18% between 2005 and 2015, and mental illness may become the leading cause of disabilities worldwide by 2030. Therefore, quality care for people with mental health problems that is provided by qualified health professionals is becoming increasingly necessary.

A professional caregiver is the person who, from different fields, is in charge of carrying out functions that exceed another person’s own abilities to take care of him/herself and to promote a better quality of life [5]. Their functions are usually related to the promotion of autonomy, mobility, hygiene, feeding, elimination (i.e., deposition) or safety, among others. In addition, they provide a point of support through functional social interaction [6]. He or she is generally trained in social health and, through a formal agreement, complies with a working schedule and receives remuneration in return [7]. Work in these types of professions, which is focused on direct contact with people with mental issues, can induce so-called burnout syndrome [8]. This concept was coined by Freudenger [9] to describe the physical and emotional exhaustion that can occur in workers in certain health institutions that tends to be associated with work characteristics such as long and variable working hours, as well as with a poor salary and high social demands, which could affect the care provided.

Some authors have begun to propose a change of focus, “offering a less pathologizing vision of caregivers, that is more focused on the analysis of those capacities and strengths that act as protective factors against prolonged stress” [10]. In a series of qualitative reports, Kramer [11] observed that attention to the positive aspects of caregivers’ work is consistent with a perspective on strengths that recognizes the capacity for continuous growth in each individual, and noted that the positive aspects of care can be important determinants of the quality of care provided. In this way, the following concepts have begun to be widely studied as fundamental characteristics of a high-quality professional caregiver: prosocial behavior, empathy, resilience and emotional intelligence.

Prosocial behavior includes specific actions of providing help or benefits to third parties in the absence of extrinsic material reward; it is the result of multiple individual and situational factors, including parenting variables and empathic traits [12]. Prosocial behavior can be postulated as a key factor in providing satisfactory care for dependent people since it shows that behaviors such as helping, sharing, consoling, caring and empathizing benefit not only the other, but also the person who performs such behaviors [13]. Another variable that might play a fundamental role in caring for people with mental illness is empathy, “which includes both a cognitive component (understanding the other person) and an emotional component (worrying about the other person)”, and which has been shown to be one of the most important predictors of prosocial behavior [12].

Resilience is defined as “a dynamic process in which psychological, social, environmental and biological factors interact to allow an individual, at any stage of life, to develop, maintain or recover their mental health despite exposure to adversity” [14] (p. 125). Factors that promote resilience include social support, the way that stressors are evaluated, and the coping style used by the caregiver. Relevant to the latter is the transactional theory of stress of Lazarus and Folkman [15], which states that the person and the environment maintain a dynamic, bidirectional, mutually reciprocal relationship, and that stress is therefore considered a process that includes the relationships between the individual and his or her environment, in which the perception of threat and/or damage causes physical and psychological reactions.

With regard to emotional intelligence, Mayer and Salovey [16] defined this concept as the ability of human beings to sense and identify emotions, understand and modify mental states. Emotional intelligence includes the mechanisms that allow the individual to manage the emotional adaptation necessary to face stressful situations. Thus, a positive attitude toward helping third parties can determine positive emotional functioning among caregivers and, in this way, help to offset the negative consequences that may arise from care [17].

Despite the recognized value of mental health and the need for care and support of people with mental illness, our understanding of the aspects that affect the quality of care to these patients is lacking. It is vitally important to study and analyze the impact that negative variables, such as burnout or perceived stress, can have on professional caregivers, and to focus on the abilities and strengths that act as protective factors to achieve improvement in the quality of life of both professionals and the patients in their charge.

Therefore, the main aim of the current study is to identify variables related to the professional care of people with mental illness (i.e., protective or stressor variables) through a systematic review.

The secondary objective is to identify the main interventions that are currently being implemented in relation to these variables for professional caregivers of people with mental illness.

## 2. Materials and Methods

This study is a systematic review of the published scientific literature regarding variables impacting the professional care of people with mental illness. The guidelines for carrying out systematic reviews proposed in the PRISMA statement were followed [18] (Appendix A). In addition, the SPIDER tool for qualitative and mixed studies was used to establish the research questions and search strategies [19]. Regarding the ethical standards, no ethical approval or participant consent is required for this type of research (i.e., systematic review).

### 2.1. Information Sources

The systematic search was performed between October and November 2019, and updated in June 2022, in the Web of Science (WoS), PubMed, ScienceDirect and Dialnet databases, including all articles published from 1900 to 2021 (inclusive). A total of 2429 articles were recovered: 135 articles from PubMed, 221 from WoS, 1637 from ScienceDirect and 436 from Dialnet.

### 2.2. Eligibility Criteria

A protocol was registered in PROSPERO and the search was conducted according to the following criteria. The identification code is CRD42022340313.

#### 2.2.1. Inclusion Criteria

The inclusion criteria were as follows: (a) articles that reported empirical research or interventions; (b) articles that referred to variables related to the professional practice of professional caregivers in the field of mental health; (c) articles in any language (to collect as many articles as possible, as well as to reduce the “Tower of Babel” effect; i.e., the prevalence of studies in a certain language over others written in lesser-represented languages [20]), and (d) articles to which full-text access was possible.

#### 2.2.2. Exclusion Criteria

The exclusion criteria were as follows: (a) articles that did not include narrative articles; (b) articles that included non-professional caregivers as participants; (c) articles that included professional caregivers who work in a field other than mental illness; (d) articles that include systematic reviews or metanalysis, and (e) articles that did not include variables related to the professional practice of care.

### 2.3. Search Strategy

The bibliographic search was carried out in three phases: an initial search to obtain an overview of the current situation, the application of inclusion and exclusion criteria, and a manual search to evaluate the results obtained. The combinations of terms used were as follows.

In PubMed, the following terms were used: “professional caregiver” AND “mental health”; “professional caregiver” AND “mental illness”; “professional care’ AND “mental health”; “professional care” AND “mental illness” in the title and abstract fields. In WoS: “professional caregiver” AND “mental health”; “professional caregiver” AND “mental illness”; “professional care” AND “mental health”; “professional care” AND “mental illness” in the topic field. In ScienceDirect: “professional caregiver” AND “mental health”; “professional caregiver” AND “mental illness”; “professional care” AND “mental health” NOT “family” NOT “relative”; “professional care” AND “mental illness”, narrowing the search to research articles and in title, abstract and keywords fields. In Dialnet: “professional caregiver” AND “mental health”; “professional caregiver” AND “mental illness”; “professional care” AND “mental health”; “professional care” AND “mental illness”.

This process was carried out by one of the authors and corroborated by another through the Covidence tool [21].

### 2.4. Data Collection

The data to be extracted from each of the instruments were also defined in advance to ensure that the information was extracted in a uniform manner. The selected documents were then recorded in a Microsoft Excel spreadsheet and in Covidence software.

Thus, the recorded information included (1) the name of the authors and the year of publication, (2) the aims, (3) the methodology used and the presence of a control group, (4) the number of participants in the sample, (5) the variables or themes included and the results obtained in each study, and (6) the limitations of each article.

### 2.5. Selection Process

The summaries of all the articles were read, and only 67 articles were considered adequate after passing an initial screening process (after eliminating 635 duplicate articles in different combinations of the various databases). After screening, an analysis of the full text of these 67 articles was carried out. As a result, 47 articles were eliminated because they included narrative articles (*n* = 5), they did not include professional caregivers in the mental health field (*n* = 13), they did not include variables related to the professional practice of care (*n* = 22), they include systematic reviews (*n* = 2) or the full text was not accessible (*n* = 5). The remaining 20 articles that met all the inclusion criteria were selected for inclusion in the systematic review. The various phases of execution of the procedure are detailed in Figure 1.

## 3. Results

The synthesis of the results of the selected studies is shown chronologically and alphabetically in Table 1. The order of analysis of the articles presented below differs from that shown in the table to facilitate the understanding of the results.

As shown in Table 1, most of the articles are research articles (*n* = 13), followed by articles dedicated to interventions (*n* = 7). In terms of the objectives of the studies included, seven investigate the variables that may affect the dispensing of care (*n* = 7) [25,28,30,33,35,38,39], seven are focused on researching or increasing the training and knowledge of professional caregivers (*n* = 7) [24,29,31,32,36,41], three are dedicated to researching the working atmosphere and how it can affect care quality (*n* = 3) [26,27,40], and three investigate the caregiver perspective (*n* = 3) [22,23,37]. The studies can be divided into quantitative (*n* = 14), qualitative (*n* = 4) and mixed methods (*n* = 2) studies, as well as into cross-sectional (*n* = 7), longitudinal (*n* = 7), inductive (*n* = 3), exploratory (*n* = 1), study case (n = 1) and other (*n* = 1) studies. In addition, four of these works do not discuss the limitations of their studies [23,30,39,41]. Through the analysis carried out, several different but interrelated topics are developed below.

Professional caregivers whose personal values are consistent with the commonly shared values of a caring profession experience less exhaustion and greater personal well-being [33]. In addition, this well-being is also implicitly and explicitly linked to personal experiences and professional practices [33], and relates to factors such as honesty, clearly defined work, competence and fulfillment of obligations [39]. In addition, the feeling of calm and a healthy perception of oneself are critical components of effective professional practice [41]. On the other hand, caregivers with psychological distress report higher burnout and compassion fatigue scores, and lower levels of compassion satisfaction [35]. Furthermore, it has been seen that with each additional year that is spent in the caregiver position, emotional tension, burnout and mental health problems increase significantly [35].

When comparing various dimensions of burnout (emotional exhaustion, depersonalization and personal fulfillment) between dyads of people with schizophrenia and depression and nurses who work as caregivers in psychiatric hospitals, there are no significant differences, and approximately a quarter of both have a high degree of exhaustion [38]. For other positive variables, such as emotional intelligence and resilience, it has been seen that both workers in this sector and subjects not working in this sector have an average level of both, so mental health professionals are no more “protected” from stressors than the rest of the population [30].

Other variables that may affect adequate care dispensing include recovery-oriented care, which is related to other variables, such as team support and interdependence, knowledge sharing, confidence, and belief in multidisciplinary collaboration, and which could be enhanced through the promotion of appropriate resources, training on best recovery practices and standardized assessment tools [25]. Job satisfaction is also an important variable in the provision of adequate care. Significant correlates of the reduction in job satisfaction include higher levels of education, increased work experience, a large number of cases, low income and few opportunities for professional development [40]. On the other hand, it has been seen that intrinsic motivational factors, such as satisfaction with the amount of responsibility, with the recognition of work, with the amount of variety in work and with the freedom of working method, lead to an increase in overall satisfaction [27]. In addition, social support plays a fundamental role in mental health-related jobs [40].

It has also been found that caregivers have a system of clearly articulated values, and twenty-one qualities can be identified that are classified into values about humans, values about disease/disability and values about the work of care, which are of critical importance for job effectiveness and satisfaction [37]. Another factor that may affect the provision of adequate care may be job satisfaction. It has been seen that it is essential that all caregivers work with a common goal, so a leader who manages the group and increases the individual skill of each worker has been proposed as a necessary figure [26]. Another aspect that plays a key role in this regard is the proximity provided by home care. This type of care is seen by professional caregivers as an opportunity to get to know the person beyond their role as a patient, which increased their involvement in the job and provided greater satisfaction than conventional care in hospitals or centers [22].

On the other hand, to reduce the negative effects that can arise as a result of the role of a professional caregiver in this area, a number of interventions have been developed in recent years. Mindfulness is used to teach participants to develop the capacity for observation, acceptance and compassion towards the emotions and thoughts that occur in the workplace. Mindfulness has been found to be useful for significant improvement in various facets of mindfulness (i.e., observational, descriptive, non-judgmental and non-reactive skills). There is also a significant increase in levels of compassion and a significant reduction in stress. Moreover, these changes have been maintained for up to 3 months after the training [28]. Another type of intervention is the Schwartz Rounds, a type of meeting in which staff discuss the emotional impact of their professional performance. In the study by Allen et al. [23], this intervention was rated as useful and beneficial by 10 professionals who perceived the rounds as an opportunity to express their feelings. This resulted in increased acceptance of their decisions, adaptive coping strategies and empathy towards and from their co-workers. They also expressed that it could help their relationship with patients. Six years later, rounds are still perceived as positive and as an opportunity to make the profession more humane.

On the other hand, the study of Barnes and Toews [41] showed that, years ago, mental health professionals had only a moderate knowledge of the principles of care, which has been seen to have important implications on the quality of care provided to patients. Another more current study yielded similar results [29]. It reported that community mental health workers caring for people with such diseases lack experience and training in this area, as only 8.7% of the participants had received community mental health training, although the post-training assessment they employed led to improvements in knowledge, especially related to home visits, case management and follow-up. According to a current study, a fundamental variable in caring for the group of people with mental illness would be continuous education and training, in addition to other aspects such as mentorship and the frequency of care [24].

In addition, individuals with certain mental illnesses, such as borderline personality disorder, often engage in countertransference toward their caregivers, who often react with avoidance and experience considerable stress [32]; as a result, many training programs are employed. Most of the work analyzed in this regard have generated benefits for participants in the form of significant increases in the competence for understanding of psychosis and relationship building [31], in the knowledge and attitudes of mental health professionals toward the recovery of people with serious mental illnesses [36], and in attitudes, self-efficacy and behavioral intent [34].

## 4. Discussion

The aim of the study is to carry out a qualitative analysis of a set of studies with different methodologies that include empirical research or interventions to identify variables affecting the professional care exercise of people with mental illness (i.e., protective or stressor variables). The secondary objective is to identify the main interventions that are currently being implemented in relation to these variables for professional caregivers of people with mental illness.

Regarding the main objective, the results of this study show that an important protective variable against caregiver burnout is the existence of coherence between the caregivers’ personal values and those values required by the profession. In this way, the actions and behaviors carried out in the workplace become meaningful for the person. There are several recent studies that confirm that congruence between the individual values of the person, and those defended by an organization are related to less emotional exhaustion and greater work commitment [42], as well as greater emotional well-being [43].

In addition, as Rose and Glass [39] state in their work, this emotional well-being would have a bilateral character since, at the same time, it is implicitly and explicitly related to personal events, as well as professional ones. This generalized interconnection was found again in a later study by the same authors in a sample of palliative care nurses [44]. Later studies, such as that of Boamah et al. [45], conducted with a sample of 3743 nurses, confirmed the importance of the relationship between these values and, in addition, it also highlighted the special relevance of the figure of a leader who unifies the group, establishes tasks with a common goal for the whole team and ensures an empowering work environment. This can help to reduce burnout, increase employee job satisfaction and improve the quality of patient care [45].

Job satisfaction also has a significant influence on the care provided to the patient, as it has been shown to have a moderating effect on the effort and eagerness dispensed. Related to this, it has also been shown that caregivers with higher satisfaction in their work environment are more likely to engage in a greater number of caregiving behaviors [46]. Other factors related to this occupational well-being are, positively, resilience developed by caregivers [47], and patients’ perception of the quality of care provided by their caregiver [48]; negatively, related factors include patients’ relapses [49]. In addition, being able to establish a closer relationship with the patient results in an increase in this job satisfaction, and at the same times prevents burnout and reduces stress. This closeness between patient and caregiver is fundamental in the patient’s recovery, as it results in more personalized care, a factor also related to the satisfaction of caregivers of people with mental illness [50,51]. Closely related to this is the concept of recovery-centered care, which, likewise, provides a more personalized and closer treatment, and would provide the same benefits for the caregiver. This term refers to a process that supports the patient’s long-term recovery efforts and that includes processes of fostering relationships, conveying hope, focusing on strengths (rather than deficits), supporting the person in engaging in purposeful activities, and educating and empowering people for self-care [52].

However, the findings of Acker’s study [40] are striking. The author states that aspects such as higher educational level and more work experience have a negative relationship with job satisfaction. A recent meta-analysis of more than 70 studies revealed that this association is established because people with a higher educational level, although they tend to benefit from aspects such as higher income, autonomy and role variety, also tend to suffer greater demands on their job in terms of working hours, task pressure, work intensity and time urgency [53]. These negative aspects would unbalance the benefits derived, a priori, from a higher educational level, and lead to higher job stress and lower job satisfaction. As for experience, it may be negatively correlated with job satisfaction due to the self-development of professionals related to caring fields, such as nursing [54]. Those who are new to their careers often experience this step as an opportunity for growth and development, something that would continue until six years of experience. After this point, professionals may begin to experience a sense of stagnation, leading to a period of disenchantment and reduced job satisfaction [54]. However, the opposite effect may occur, since job satisfaction may start to increase after 15 years of experience due to a reconciliation between the personal and professional development of professionals, and because those who continue to maintain a lower level of satisfaction tend to change careers [54].

Regarding the training of professionals about mental illness, comparing studies such as the Barnes and Toews study from almost 20 years ago [41], it can be seen that the picture is still similar. Several articles reveal a lack of knowledge and experience in this regard [29,55,56]. Apparently, this lack of staff training would be perceived as a key challenge in providing good quality care to patients with mental illness [55]. There are several elements that would play a key role in providing higher quality care to people with mental illness by professional caregivers [55]: increased knowledge about symptoms and diagnoses, increased communication and interactions with the patient, and a reduction in the stigma of mental illness.

Another aspect related to the main objective is the identification of the variables that act as stressors in the professional care practice of people with mental illness. Results show that with each additional year spent as a professional caregiver, emotional stress, burnout and mental health problems increase significantly. A systematic review by the team of O’Connor et al. [57] shows that mental health professionals have high levels of emotional exhaustion and moderate depersonalization. In addition, staff working in community mental health teams may be more vulnerable to burnout than those working in specialized teams [57]. On the other hand, a study conducted with a sample of 420 healthcare professionals analyzed the different dimensions of burnout in different time periods [58]. The results show a significant increase in emotional exhaustion and depersonalization over time. This shows that the demands and conditions of work environments can become stressors that can also increase the emotional exhaustion of workers chronically if they persist over time.

In addition, this emotional exhaustion can have a negative impact on patient care, as it has been shown to affect performance. In recent years, numerous mindfulness programs have been developed to alleviate this exhaustion in workers; however, the results do not show a high level of effectiveness. A systematic review [59] of thirty-four studies using this type of program also showed a low level of evidence. Only four of the interventions showed a significant improvement in burnout; however, this may be due to the fact that most of the studies had numerous methodological limitations. Other types of interventions that have been found to improve the quality of life of workers in this type of environment are those that promote open discussion, as they allow for the provision of mutual support among peers and result in decreased feelings of isolation and greater comfort in sharing social and emotional experiences [60]. These results are consistent with those found in a systematic review conducted in 2019 [61], which reveals that the most commonly used interventions in these types of professions that tend to have a positive effect on reducing burnout are those that promote communication skills, teamwork, and participatory programs.

This work has some limitations. These include a possible publication bias (i.e., the tendency to publish research with significant results and not those without significant differences), which is characteristic of review-type studies. This, added to the fact that the review was performed only on peer-reviewed publications, may have contributed to a possible loss of information or another line of argument. On the other hand, there is a high degree of methodological heterogeneity among the articles included in terms of the characteristics of the study.

On the other hand, the article has several strengths. First, four different databases have been used, which has allowed greater coverage of the results. In addition, one of them is WoS, the oldest, most widely used and authoritative database of research publications and citations in the world [62], which allows greater efficiency in obtaining greater coverage of articles [63], and which has a human team that avoids total automation of the process [64]. In addition, the study tried to minimize linguistic bias or the “Tower of Babel” effect [20], by searching for articles presented in any language, to avoid including only those studies published in English. In the future, this article could be enriched by a manual search for additional articles, for example, in the references of other articles or in the gray literature. Moreover, both the search process and the data extraction process should be performed and corroborated by more than one researcher. It is also necessary to develop a protocol for recording inclusion and exclusion criteria for primary studies.

## 5. Conclusions

After integrating and analyzing the results, it can be concluded that there are multiple factors of various kinds that can influence the quality of care provided by professional caregivers to persons with mental illness. Predictors of the quality of care provided include job satisfaction, congruence between personal values and those advocated by the organization where they work, and working with a common goal in an environment that facilitates employee empowerment.

On the other hand, it has also been seen that professional caregivers face several stressors in their work environment: working in the same job for consecutive years, working in community mental health teams and burnout. These stressors are related to increased emotional stress, mental health problems, greater vulnerability to burnout and poorer performance and quality of care.

Likewise, several studies report that there is a clear lack of training and experience on the part of caregivers about mental illness and how to provide adequate care. In this regard, promoting knowledge of the symptoms and the disease, facilitating communication and interactions with the patient and reducing the stigma of mental illness among workers can play a key role.

This study identifies different variables related to professional care for people with mental illness. However, it is not focused exclusively on those centered on a pathologizing view of caregiving, but also on variables that promote the well-being of the caregiver and, as a consequence, of the person being cared for, and that can act as protective factors. In this way, the aim is to raise awareness among researchers of the importance of changing the point of view and the focus of research towards a more positive and strengths-based approach of the caregiving profession. On a practical level, it is worth stressing the importance of carrying out training programs and specific training of caregivers towards mental illness, since several studies have highlighted the lack of knowledge and stigma that still prevails among these professionals towards this condition.

Research and clinical practices related to caregivers, both formal and informal, are of vital importance in all societies today, since it is predicted that mental illness may become the leading cause of disability worldwide by 2030 [4]. However, they are also necessary because of the rise of other factors such as the aging of the population, which is expected to double in the next 30 years to 1.5 billion people by 2050 [65]. It is therefore vitally important to care for those who care, because only in this way will it be possible to provide quality care for each and every person who requires it.

## Figures and Tables

**Figure 1 healthcare-10-01225-f001:**
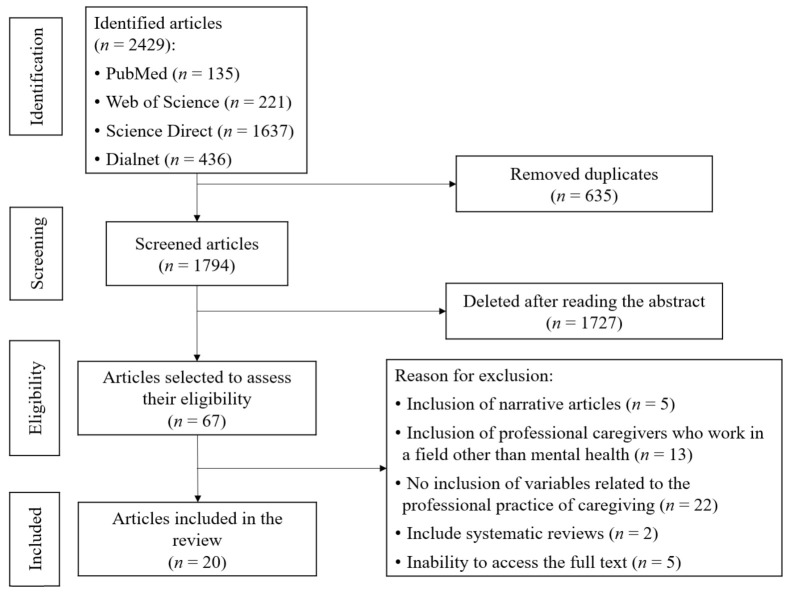
PRISMA flowchart.

**Table 1 healthcare-10-01225-t001:** Synthesis of the included articles.

Authors and Year	Aims	Methodology and Presence of Control Group	Participants and Country	Variables or Themes and Results	Limitations
Giménez-Díez et al., (2021) [22]	Explore nurses’ perceptions and constructions about care in crisis resolution home treatment teams (CRHTT) services	Case study Qualitative (semi-structured interviews) No control group	10 nurses who had worked or were working in CRHTT, Spain	Nurses’ perspectives of the care provided, nursing setting of care at home and nursing care plan at home; nurses believed that providing home care facilitates an intimate perspective, which creates a special bond with patients and instils personal satisfaction with their work; nurses felt more involved and responsible when they were close to patients and applied care adapted to real needs, often establishing a close relationship with the patient.	The study explored mental health nursing experiences in a specific setting; it may have been appropriate to conduct a focus group to gain feedback on the participants’ initial analysis; qualitative studies have limited data extrapolation; the sample size can be considered insufficient or biased; this study is difficult to replicate in other contexts.
Allen et al., (2020) [23]	Evaluate mental health professionals experience of the rounds using a mixed-methods approach comprising data collection through standardized evaluation forms, focus groups, and facilitator notes taken during the rounds	Long-term study Quantitative (standardized evaluation) and qualitative (focus groups and round facilitators notes) No control group	150 mental health professionals, United Kingdom	Relevance of the rounds to the participants’ work, expression of emotions, sharing similar emotions and experiences, feelings of guilt; the rounds were rated as helpful, insightful and relevant. Participants commented that the rounds had helped them feel able to express both negative and positive feelings they had towards users, and this was considered beneficial for themselves and for their relationships with their patients	None described
Avery et al., (2020) [24]	Explore characteristics of variables (personal, educational and professional) more frequently associated with and more predictive of nursing preparedness	Descriptive correlational design Quantitative (scales and surveys) No control group	260 nurses from a tertiary health system, United States	Characteristics of variables (personal, educational and professional) associated with preparedness; the three characteristics of professional experiences that best prepare a nurse to care for this population are mentorship, frequency of care and continuing education.	Findings were dependent on perceptions of participants as opposed to observed or measured data; participant responses were aggregated, therefore, determination of response variation from nurses employed at small versus large or urban versus rural hospitals was not possible.
Fleury et al., (2018) [25]	Identify variables associated with perceived recovery-oriented care among mental health professionals	Cross-sectional study Quantitative (scales and questionnaires) No control group	315 mental health professionals and 41 managers of service networks, Canada	Recovery-oriented care, team support, team autonomy, involvement in decisions, team reflexivity, team conflict, team collaboration, job satisfaction, trust, team climate; work in primary care or outpatient mental health services, team support, knowledge-sharing, team reflexivity, trust, belief in multidisciplinary collaboration and frequency of interaction with other organizations are significantly and positively related to recovery-oriented care.	Impossibility of making causal inferences due to cross-sectional design; no links established between recovery-oriented care and patient outcomes in terms of personal recovery; results based on only four regions of Quebec.
Pileño et al., (2018) [26]	Analyze the organizational culture of the team of professionals working in the mental health network	Descriptive, inductive study Qualitative (in-depth interview and focused interview) No control group	55 mental health professionals, Spain	Main theme: the team. Five subthemes: (1) getting along on the unit; (2) getting along with patients; (3) personal resources for dealing with patients; (4) adaptive resources of team members; (5) team resources.	Inability to obtain access to a hospital and lack of cooperation from certain staff members when participating in in-depth interviews.
Goetz et al., (2017) [27]	Evaluate aspects of job satisfaction and the work atmosphere of mental health professionals who work in the comprehensive care model and explore associations between satisfaction with different aspects of their work, individual characteristics, work atmosphere, and general job satisfaction	Exploratory study Quantitative (scales) No control group	321 community mental health professionals, Germany	Job satisfaction and working atmosphere; intrinsic motivational elements such as satisfaction with the amount of responsibility, with job recognition, with the amount of variety at work, and with freedom of working method increased overall job satisfaction.	Possible selection bias
Suyi et al., (2017) [28]	Examine the effectiveness of a mindfulness program in increasing mindfulness and compassion and reducing stress and exhaustion, among mental health professionals	Non-experimental design, pre- and post-testing with follow-up Quantitative (scales and questionnaires) No control group	37 professionals working at a mental health institute, Singapore	Mindfulness, compassion, stress and burnout; significant improvement in four of the five mindfulness facets and in compassion levels, and a significant reduction in stress following intervention, but no change was observed for burnout.	Small sample size; participants from the same institution; lack of control group; experimental and social desirability bias (the researcher was the instructor for the program); study not generalizable to other health professionals.
Yang et al., (2017) [29]	Provide an interdisciplinary community mental health training program and assess the effect of training on staff knowledge of mental health and confidence in their roles	Group design with pre- and post-testing Quantitative (scales and questionnaires) No control group	48 mental health professionals, China	Community mental health knowledge and confidence in managing people with mental health issues; the score on every item, except the item on empathy and the total/average score, was significantly increased.	Non-objective measure of knowledge improvement; transfer of learning to the workplace was not measured.
Frajo-Apor et al., (2015) [30]	Investigate emotional intelligence and resilience in mental health professionals compared to a control group who did not work in healthcare	Cross-sectional design Quantitative (test and scales) With control group	61 mental health professionals and 61 participants working in unrelated areas, Austria	Emotional intelligence and resilience; the two groups did not differ significantly from each other, neither in terms of emotional intelligence nor resilience; positive correlation between emotional intelligence and resilience; mental health professionals were not more resilient than the general population.	None described
Sørlie et al., (2015) [31]	Increase skills, joint understanding, and collaboration in working with people with severe mental illness	Prospective study of longitudinal cohort Quantitative (scales and questionnaires) No control group	1258 professionals working in different services related to mental health, Norway	Understanding psychosis, building relationships, using own reactions, multidisciplinary collaboration, teamwork and collaboration and supporting relatives; significant increase in participants’ experienced competence in all variables, especially for the understanding of psychosis and relationship building; no significant variance at the program level.	The study focused solely on the changes in competence experienced by the participants, and not on whether patients experienced an improvement in services; data used was collected between 1999 and 2005.
Utrera et al., (2014) [32]	Evaluate the effectiveness of a training program in emotional intelligence for levels of satisfaction, emotional intelligence and stress in nurses treating patients diagnosed with borderline personality disorder	Quasi-experimental, prospective longitudinal design Quantitative (scales and inventories) No control group	77 nurses in a mental health unit, Spain	-	Possible social desirability bias (participants’ responses aimed at giving a good image of themselves); possible learning bias (repeated use of same measurement instrument); limited ability to generalize results.
Veage et al., (2014) [33]	Explore the life values of mental health professionals, their personal values relating to work, and the links between these values and well-being and exhaustion	Correlational study Quantitative (scales and inventories) No control group	106 mental health professionals working for nongovernmental organizations, Australia	Burnout, psychological well-being, personal life values and personal values related to work; congruence between life values and personal work-related values was related to greater well-being and less burnout; honesty, clearly defined work, competence, and fulfilment of obligations were associated with less exhaustion and greater well-being.	Results not generalizable to other professions; inability to determine the causal direction
Irvine et al., (2012) [34]	Evaluate an internet-based training program on mental illness for nursing assistants, and explore its effects and acceptance in health professionals	Randomized treatment/control pre-post design for nursing assistants; quasi-experimental pre-post design for health professionals Quantitative (scales and interviews) With control group	70 nursing assistants and 16 health professionals, USA	Knowledge, self-efficacy, knowledge of myths versus facts, attitudes, self-efficacy and behavioral intentions; significant and medium-to-large effects were obtained on five of the six outcome measures (except self-efficacy) for nurse aides; significant effects on five of six outcome measures (except myths), with medium-large effect sizes.	Need for follow-up evaluations, preferably with in vivo evaluation; impossibility of verifying selection criteria; small sample size.
Rossi et al., (2012) [35]	Evaluate exhaustion, compassion fatigue, and satisfaction with compassion among community mental health services staff	Cross-sectional design Quantitative (scales and questionnaires) No control group	260 community mental health service professionals, Italy	Burnout, compassion fatigue and compassion satisfaction; distressed workers had a mean value of compassion satisfaction significantly lower than the nondistressed workers; workers with psychological distress reported both higher burnout and compassion fatigue scores; significant increase in the burnout and compassion fatigue scores was also detected for each additional year spent.	Impossibility of determining causality; potentially significant variables not included; possible type II errors due to small sample size.
Wilrycx et al., (2012) [36]	To investigate the effectiveness of a recovery-oriented training program on the knowledge and attitudes of mental health professionals about the recovery of people with severe mental illness	Two-group multiple intervention interrupted time series design (a variant of the staggered wedge test design) Quantitative (questionnaires and inventories) No control group	210 mental health professionals, Netherlands	Recovery knowledge and knowledge attitudes; significant increase in both variables.	No reference data to compare; absence of data from psychosocial studies; too many measurement points made it difficult to maintain cooperation and motivation of the mental health professionals.
Piat et al., (2007) [37]	Examine caregivers’ and residents’ perspectives on the support relationship in adult care homes	Inductively focused design within a naturalistic paradigm Qualitative (semi-structured interviews) No control group	20 caregivers in care homes, Canada	Ten themes: (1) the qualities and skills of caregivers; (2) how caregivers learned their job; (3) perceived difficulties, needs and expectations of residents; (4) goals in caring for residents; (5) approaches to helping; (6) caregiver–resident relationships; (7) caregiver–professional relationships; (8) differences between caregiving and professional helping; (9) caregivers’ time allocation between work, family and social life; and (10) the advantages and disadvantages of caregiving.	The sample was not representative of all caregivers in care homes; small sample; possible social desirability bias; need for comparative studies between formal and informal caregivers.
Angermeyer et al., (2006) [38]	Examine the similarities and differences between levels of exhaustion in family members and nurses caring for patients with mental illness	Cross-sectional design Quantitative (inventories and scales) No control group	94 partners of people with depression, 39 partners of people with schizophrenia, and 128 health professionals in a psychiatric hospital, Germany	Burnout; about one fourth of the respondents in both groups showed a high degree of burnout, but no significant differences were found in the three dimensions of burnout (emotional exhaustion, depersonalization, and personal accomplishment) for the two groups of caregivers.	Low response rate; only partners of people with schizophrenia and depression were interviewed; the results might not be generalizable beyond Germany.
Rose and Glass (2006) [39]	Examine the degree of emotional well-being in community mental health nurses and identify factors that impact their professional practice	Descriptive, inductive design Qualitative (interviews) No control group	5 nurses in community mental health centers, Australia	Three themes: (1) being able to speak out (or not); (2) being autonomous (or not); (3) being satisfied (or not).	None described
Acker (2004) [40]	Examine the relationship between the organizational conditions of mental health agency workers and their job satisfaction	Cross-sectional design Quantitative (scales) No control group	259 professionals working for mental health agencies, USA	Role conflict, role ambiguity, social support, extent of opportunities for professional development, type of work activities, job satisfaction and intention to leave; both role conflict and role ambiguity had statistically significant negative correlations with job satisfaction and positive correlations with intention to leave; social support had statistically significant positive correlations with job satisfaction and negative correlations with intention to leave; opportunities for professional development were positively correlated with job satisfaction and negatively correlated with intention to leave; role conflict also had statistically significant negative correlations with social support.	Possible influence situational state of mind when responding about job satisfaction.
Barnes and Toews (1985) [41]	Examine the knowledge of mental health workers about the principles of care for chronic mental illness	Cross-sectional design Quantitative (test) No control group	246 professionals working for mental health associations, Canada	Knowledge in the field of caring for chronic mental disorders; mental health professionals were moderately knowledgeable on this topic (mean score 66%); errors made were more commonly in the direction of overenthusiastic support for the community approach; there were no differences in knowledge scores by a demographic or professional status variables included in this study.	None described

## Data Availability

The raw data supporting the conclusions of this article will be made available by the authors, without undue reservation.

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
