# Peer review of "Variables Impacting the Quality of Care Provided by Professional Caregivers for People with Mental Illness: A Systematic Review"

_healthcare, 2022, doi:10.3390/healthcare10071225_

Round 1
Reviewer 1 Report
Thank you for giving me the opportunity to review this manuscript. This is a systematic review that is focused on identifying variables (stressors and protective factors) impacting the quality of care provided by professional caregivers to patients with mental illness.
Overall, the studies that were included were sufficient to highlight that and the manuscript was written well. However, a few points must be considered by the authors before considering it for publication.
1. Title should be reward like: "Variables impacting the quality of care provided by professional caregivers for people with mental illness: A systematic review"
2. Introduction:
-Sentence in line 56: better to be the last sentence of the above paragraph.
-Sentence in line 57: start of new paragraph
-Remove sentences in lines 104-109. Nothing in the provided introduction discussed the patients or caregivers rights and stigma that they may get exposed to, although this could exist, hence no support of such claims. Instead use this "Despite the recognized value of mental health and the need for care and support of people with mental illness, our understanding to the aspects that affect the quality of care to these patients is lacking. It is vitally important....."
-Remove lines 121 to 125. It is a repetition of what has been written in 114-120
3. Methods
-Line138: Research was done in 2019. Authors have to run the research again as it has been approximately 3 years since their literature research.
-Line 148: How to include systematic reviews in this systematic review? These reviews must be excluded.
-Line 149: use professional instead of formal
-Line 156: use non-professional instead of informal
-Figure 1 is quite confusing and doesn't reflect the text in selection process, particularly in the identification and screening step. Figure 1 should start with 2403--> remove duplicate (n=12) to be 2391---> delete after reading (n=2326) to be 65 ---> remove after screening (n=45) to be 20 to be included in this review
3. Results:
-It is recommended that studies in table 1 are arranged according to their type and strength of evidence, such that RCTs appear first, then prospective studies, retrospective studies, cross-sectional, case report...etc.
-Also, systematic reviews must be excluded as you can not include them in this systematic review.
-Line 201: Two are dedicated....(and n=3??) need to be corrected
4. Discussion:
-Need to discuss how higher education and increased work experience (stated in line 261) correlate with reduction in job satisfaction. Does that mean they are stressors for professional caregivers? Such claim has to supported strongly and discussed very well to convince the readers.
-Line 293: How well being that comes with professional experience act as protective factor from burnout? Authors previously stated (line 261) that professional experience reduces job satisfaction, Is it a stressor? Need to avoid such contradictions that confuse the readers.
-Line 335: What are the interventions that showed significant improvement in burnout? Stating that will add more value to the review.
-Line 342: authors need to provide brief explanation about recovery-focused care
Author Response
June 23, 2022, Mexico City
Responses to reviewers
Variables impacting the quality of care provided by professional caregivers for people with mental illness: A systematic review
Prof. Dr. Paul B. Tchounwou
Editor-in-Chief
Thank you for the opportunity to resubmit our manuscript ID: healthcare-1765988 " Variables impacting the quality of care provided by professional caregivers for people with mental illness: A systematic review " to Healthcare. We appreciate the opportunity to publish in Healthcare. We have carefully reviewed the reviewers' valuable comments. To ensure that we have fully addressed all of your concerns, we have revised our manuscript based on the reviewers' suggestions.
Below, we include our point-by-point responses to the reviewers' corrections and comments and describe in detail the changes made to the manuscript.
Thank you again for the opportunity to resubmit our manuscript. I appreciate any help you can provide.
Sincerely,
Filiberto Toledano-Toledano, Ph.D.
Federico Gómez Children’s Hospital of Mexico, National Institute of Health.
Dr. Márquez 162, Doctores, Cuauhtémoc, México City, 06720, México.
+ 52 55 52289917, ext. 4318. E-mail: filiberto.toledano.phd@gmail.com
Reviewer 1
Comments and Suggestions for Authors
General comment
Thank you for giving me the opportunity to review this manuscript. This is a systematic review that is focused on identifying variables (stressors and protective factors) impacting the quality of care provided by professional caregivers to patients with mental illness.
Overall, the studies that were included were sufficient to highlight that and the manuscript was written well. However, a few points must be considered by the authors before considering it for publication.
Response:
Thank you very much for reading our manuscript and for providing your help and recommendations. We feel that your comments have greatly improved our study.
Change:
None.
- Title should be reward like: "Variables impacting the quality of care provided by professional caregivers for people with mental illness: A systematic review".
Response:
The title has been modified as you indicated.
Change:
“Variables impacting the quality of care provided by professional caregivers for people with mental illness: A systematic review.”
- Introduction:
-Sentence in line 56: better to be the last sentence of the above paragraph.
Response:
The sentence has been added to the previous paragraph.
Change:
“…, and mental illness may become the leading cause of disabilities worldwide by 2030. Therefore, quality care for people with mental health problems that is provided by qualified health professionals is increasingly necessary.”
-Sentence in line 57: start of new paragraph.
Response:
The sentence has been separated from the previous one and starts the paragraph.
Change:
“A professional caregiver is…”
-Remove sentences in lines 104-109. Nothing in the provided introduction discussed the patients or caregivers rights and stigma that they may get exposed to, although this could exist, hence no support of such claims. Instead use this "Despite the recognized value of mental health and the need for care and support of people with mental illness, our understanding to the aspects that affect the quality of care to these patients is lacking. It is vitally important....."
Response:
Thanks for the recommendation, the sentence has been replaced.
Change:
“Despite the recognized value of mental health and the need for care and support of people with mental illness, our understanding to the aspects that affect the quality of care to these patients is lacking.”
-Remove lines 121 to 125. It is a repetition of what has been written in 114-120.
Response:
Thanks, the sentences have been deleted.
Change:
Sentences deleted.
- Methods
-Line138: Research was done in 2019. Authors have to run the research again as it has been approximately 3 years since their literature research.
Response:
Thank you for the recommendation. As you can see from the methodology, results and discussion, the review has been updated to include the literature published on the subject in these years.
Change:
New information with the three new articles in methodology, results and discussion.
-Line 148: How to include systematic reviews in this systematic review? These reviews must be excluded.
Response:
Systematic reviews have been eliminated from the review.
Change:
The systematic reviews included a priori in the review have been eliminated.
-Line 149: use professional instead of formal.
Response:
The word has been substituted.
Change:
“Articles that referred to variables related to the professional practice of formal profes-sional caregivers…”
-Line 156: use non-professional instead of informal.
Response:
Thanks, the word has been substituted.
Change:
“Articles that included non-professional caregivers as participants”
-Figure 1 is quite confusing and doesn't reflect the text in selection process, particularly in the identification and screening step. Figure 1 should start with 2403--> remove duplicate (n=12) to be 2391---> delete after reading (n=2326) to be 65 ---> remove after screening (n=45) to be 20 to be included in this review.
Response:
The figure has been modified for better understanding and contains the new searches.
Change:
- Results:
-It is recommended that studies in table 1 are arranged according to their type and strength of evidence, such that RCTs appear first, then prospective studies, retrospective studies, cross-sectional, case report...etc.
Response:
We really appreciate your recommendation; it seems to us a very interesting way to order the articles. However, in our experience we believe that in an article like this it is better to order them chronologically in order to see the progression of interventions and research. But we take it into account for future research.
Change:
None.
-Also, systematic reviews must be excluded as you can not include them in this systematic review.
Response:
Systematic reviews have been eliminated from the review.
Change:
The systematic reviews included a priori in the review have been eliminated.
-Line 201: Two are dedicated....(and n=3??) need to be corrected
Response:
Thank you, the error has been corrected.
Change:
“Three are dedicated to researching…”
- Discussion:
-Need to discuss how higher education and increased work experience (stated in line 261) correlate with reduction in job satisfaction. Does that mean they are stressors for professional caregivers? Such claim has to supported strongly and discussed very well to convince the readers.
Response:
Thank you, both variables have been treated and discussed in greater depth. We believe that thanks to your recommendation, this has provided a more solid basis for discussion.
Change:
“However, the findings of Acker's study [40] are striking. The author states that aspects such as higher educational level and more work experience have a negative relationship with job satisfaction. A recent meta-analysis of more than 70 studies revealed that this association is established because people with a higher educational level, although they tend to benefit from aspects such as higher income, autonomy and role variety, also tend to suffer greater demands on their job in terms of working hours, task pressure, work intensity and time urgency [53]. These negative aspects would un-balance the benefits derived, a priori, from a higher educational level, and lead to higher job stress and lower job satisfaction. As for experience, it may be negatively correlated with job satisfaction due to the self-development of professionals related to caring fields, such as nursing [54]. Those who are new to their careers often experience this step as an opportunity for growth and development, something that would continue until six years of experience. After this point, professionals may begin to experience a sense of stagnation, leading to a period of disenchantment and reduced job satisfaction [54]. However, the opposite effect may occur, since job satisfaction may start to increase after 15 years of experience due to a reconciliation between the personal and professional development of professionals and because those who continue to maintain a lower level of satisfaction tend to change careers [54].”
-Line 293: How well being that comes with professional experience act as protective factor from burnout? Authors previously stated (line 261) that professional experience reduces job satisfaction, Is it a stressor? Need to avoid such contradictions that confuse the readers.
Response:
Sorry, perhaps this is not well expressed. We are referring rather to personal and professional events, not to work experience maintained over the years. We have rewritten the paragraph to make it easier to understand.
Change:
“In addition, as Rose and Glass [39] state in their work, this emotional well-being would have a bilateral character since, at the same time, it is implicitly and explicitly related to personal events, as well as professional ones.”
-Line 335: What are the interventions that showed significant improvement in burnout? Stating that will add more value to the review.
Response:
Thank you for your comment, we have made it more explicit in the results and added new information in the discussion.
Change:
“On the other hand, to reduce the negative effects that can arise as a result of the role of professional caregiver in this area, a number of interventions have been devel-oped in recent years. Mindfulness is used to teach participants to develop the capacity for observation, acceptance and compassion towards the emotions and thoughts that occur in the workplace. Mindfulness has been found to be useful for significant im-provement in various facets of mindfulness (i.e., observational, descriptive, non-judgmental and non-reactive skills). There is also a significant increase in levels of compassion and a significant reduction in stress. Moreover, these changes have been maintained for up to 3 months after the training [28]. Another type of intervention is the Schwartz Rounds, a type of meeting in which staff discuss the emotional impact of their professional performance. In the study by Allen et al. [23], this intervention was rated as useful and beneficial by 10 professionals who perceived the Rounds as an op-portunity to express their feelings. This resulted in increased acceptance of their deci-sions, adaptive coping strategies and empathy towards and from their co-workers. They also expressed that it could help their relationship with patients. Six years later, Rounds was still perceived as positive and as an opportunity to make the profession more humane.”
“Other types of interventions that have been found to improve the quality of life of workers in this type of environment are those that promote open discussion, as they allow for the provision of mutual support among peers and result in decreased feelings of isolation and greater comfort in sharing social and emotional experiences [60]. These results are consistent with those found in a systematic review conducted in 2019 [61], which reveals that the most commonly used interventions in these types of professions that tend to have a positive effect on reducing burnout are those that promote communication skills, teamwork, and participatory programs.”
-Line 342: authors need to provide brief explanation about recovery-focused care.
Response:
We have added an explanation of what this concept is.
Change:
“Closely related to this is the concept of recovery-centered care, which likewise provides a more personalized and closer treatment and would provide the same benefits for the caregiver. This term refers to a process that supports the patient's long-term recovery efforts and that includes processes of fostering relationships, conveying hope, focusing on strengths (rather than deficits), supporting the person in engaging in purposeful activities, and educating and empowering people for self-care [52].”
Reviewer 2 Report
Thank you for the opportunity to read and review this paper. This paper aims to identify stressors and protective factors for those working in mental health care through a review of the peer reviewed published literature on the topic. Protective factors are particularly emphasized with a view to a strengths based approach to supporting care providers in their work.
The article is well organized and well written. I hesitate to comment on the strength of the method as I have not completed such a review myself, so lack expertise. The reporting of findings is laid out in an understandable way, and the discussion that follows derives from the findings. There are a few times when it is unclear whether the authors are writing about factors affecting the mental health and capacity of healthcare workers, or about their clients – generally I was able to understand after re-reading. There is often an overlap between what stresses a client and the stress experienced by the worker.
This review spans articles published over a significant stretch of time (back to 1985) but mostly includes recent articles that were available. (I do wonder how relevant Barnes and Toews’ findings on MH professionals’ knowledge from 1985 are today??) I do not have substantial suggestions for improvement.
Author Response
June 23, 2022, Mexico City
Responses to reviewers
Variables impacting the quality of care provided by professional caregivers for people with mental illness: A systematic review
Prof. Dr. Paul B. Tchounwou
Editor-in-Chief
Thank you for the opportunity to resubmit our manuscript ID: healthcare-1765988 " Variables impacting the quality of care provided by professional caregivers for people with mental illness: A systematic review " to Healthcare. We appreciate the opportunity to publish in Healthcare. We have carefully reviewed the reviewers' valuable comments. To ensure that we have fully addressed all of your concerns, we have revised our manuscript based on the reviewers' suggestions.
Below, we include our point-by-point responses to the reviewers' corrections and comments and describe in detail the changes made to the manuscript.
Thank you again for the opportunity to resubmit our manuscript. I appreciate any help you can provide.
Sincerely,
Filiberto Toledano-Toledano, Ph.D.
Federico Gómez Children’s Hospital of Mexico, National Institute of Health.
Dr. Márquez 162, Doctores, Cuauhtémoc, México City, 06720, México.
+ 52 55 52289917, ext. 4318. E-mail: filiberto.toledano.phd@gmail.com
Reviewer 2
Comments and Suggestions for Authors
General comment
Thank you for the opportunity to read and review this paper. This paper aims to identify stressors and protective factors for those working in mental health care through a review of the peer reviewed published literature on the topic. Protective factors are particularly emphasized with a view to a strengths based approach to supporting care providers in their work.
The article is well organized and well written. I hesitate to comment on the strength of the method as I have not completed such a review myself, so lack expertise. The reporting of findings is laid out in an understandable way, and the discussion that follows derives from the findings. There are a few times when it is unclear whether the authors are writing about factors affecting the mental health and capacity of healthcare workers, or about their clients – generally I was able to understand after re-reading. There is often an overlap between what stresses a client and the stress experienced by the worker.
This review spans articles published over a significant stretch of time (back to 1985) but mostly includes recent articles that were available. (I do wonder how relevant Barnes and Toews’ findings on MH professionals’ knowledge from 1985 are today??) I do not have substantial suggestions for improvement.
Response:
Thank you very much for your comments. It is true that there is a fine line between the well-being of caregivers and the dispensing of care and, consequently, toward the well-being of the people receiving care. But in general, the manuscript is focused on the group of professionals. Thank you for appreciating our strengths and for providing us with some areas for improvement.
As for the 1985 Barnes and Toews article, thanks to your commentary, we have made explicit a comparison between the training of professionals almost 30 years ago and now, which has made clear the need to continue working on the training of professionals to fill this gap that has been maintained for so many years.
Change:
“On the other hand, the study of Barnes and Toews [41] showed that, years ago, mental health professionals have had only a moderate knowledge of the principles of care, which has been seen to have important implications on the quality of care provided to patients. Another more current study yielded similar results [29]. It reported that…”
“Regarding the training of professionals about mental illness, comparing studies such as the Barnes and Toews study from almost 20 years ago [41], it can be seen that the picture is still similar. Several articles reveal a lack of knowledge…”
Reviewer 3 Report
Thank you for submitting your manuscript. The research seems very interesting to me but I consider that the manuscript should improve some points:
-Overview
The bibliographic review reaches until 2019. It is recommended to update the research with some more recent contributions in the discussion section.
Citation style varies in the manuscript. Sometimes only the number appears between [ ], in others the last name and the number. Please review to homogenize.
The writing style should avoid personal connotations. Check "we", "our".
- Introduction
line. 39 Initially, "dependency" is defined as a deficit in bodily function that leads to physical limitation and, later, it is linked to mental health. This union I consider is not clear enough.
line. 42 The manuscript goes so far as to state that "many people with mental illness are a part of this group". Given this, what part of the group of dependent people are you referring to? And, if only some people with mental problems, which ones?
line. 57 The manuscript defines a professional caregiver as the person who attends to demands for care and people in situations of dependency. I consider this definition limited, the professional caregiver has many areas, attends to needs that may not be expressed in demands and to people who may not have dependency. In any case, professional care is not linked to the Nursing profession, which is the discipline responsible for the study of care. Authors are requested to review the definition and justify this limitation.
- Aim
line.114 The main aim includes a methodological aspect. The aim is "to identify the variables...". The qualitative analysis and systematic review should not be included in this section.
-Method
2.1 Information sources
The study process with the phases carried out is not an information resource. This information would be more correct in section 2.3.
- Discussion
line. 281 Clarify the concept of "mixed methods studies" because there is a type of mixed methodology that is not applied in the study. The idea can be "a set of studies with different methodologies" or something similar that the authors consider.
line. 352 The manuscript states that only one researcher performs the review. This is an important limitation in the methodology. In addition, in the contribution of authors it seems that there are many people who participate in each phase of the study. Please, the authors should clarify who performs which phase.
line. 360 The manuscript states that it collects articles from "any language" but it seems that only English and Spanish are included. The authors must be rigorous in the analysis because studies in German, Chinese, Portuguese, or other languages are not included.
-Conclusions
The section repeats a lot of what was stated in the discussion. The authors must review it and guide it to a conclusion related to the aim, the academic and clinical impact of the research, and even future lines.
I hope that these considerations can be of help in revising the manuscript and resubmitting it.
Thank you very much
Author Response
June 23, 2022, Mexico City
Responses to reviewers
Variables impacting the quality of care provided by professional caregivers for people with mental illness: A systematic review
Prof. Dr. Paul B. Tchounwou
Editor-in-Chief
Thank you for the opportunity to resubmit our manuscript ID: healthcare-1765988 " Variables impacting the quality of care provided by professional caregivers for people with mental illness: A systematic review " to Healthcare. We appreciate the opportunity to publish in Healthcare. We have carefully reviewed the reviewers' valuable comments. To ensure that we have fully addressed all of your concerns, we have revised our manuscript based on the reviewers' suggestions.
Below, we include our point-by-point responses to the reviewers' corrections and comments and describe in detail the changes made to the manuscript.
Thank you again for the opportunity to resubmit our manuscript. I appreciate any help you can provide.
Sincerely,
Filiberto Toledano-Toledano, Ph.D.
Federico Gómez Children’s Hospital of Mexico, National Institute of Health.
Dr. Márquez 162, Doctores, Cuauhtémoc, México City, 06720, México.
+ 52 55 52289917, ext. 4318. E-mail: filiberto.toledano.phd@gmail.com
Reviewer 3
Comments and Suggestions for Authors
General comment
Thank you for submitting your manuscript. The research seems very interesting to me but I consider that the manuscript should improve some points:
Response:
Thank you very much for reading and reviewing our manuscript. They have undoubtedly made it much better.
Change:
None.
-Overview
The bibliographic review reaches until 2019. It is recommended to update the research with some more recent contributions in the discussion section.
Response:
Thank you for the recommendation. As you can see from the methodology, results and discussion, the review has been updated to include the literature published on the subject in these years.
Change:
New information with the three new articles in methodology, results and discussion.
Citation style varies in the manuscript. Sometimes only the number appears between [ ], in others the last name and the number. Please review to homogenize.
Response:
Thank you, we have reviewed all the citations and have homogenized them so that they appear in the same format.
Change:
Citations have been reviewed, corrected and ordered.
The writing style should avoid personal connotations. Check "we", "our".
Response:
Thank you, the personal connotations have been removed.
Change:
Personal connotations have been replaced by impersonal tenses.
- Introduction
line. 39 Initially, "dependency" is defined as a deficit in bodily function that leads to physical limitation and, later, it is linked to mental health. This union I consider is not clear enough.
Response:
We have modified the first two paragraphs to clarify the link between dependence and mental illness. Thanks to your comment we believe it is now better understood.
Change:
“According to the International Classification of Functioning, Disability and Health (ICF), dependency is defined as the result of a deficit in bodily function (i.e., physiological functions of the body systems, including psychological functions), bringing with it a limitation in activity”
“There are several causes that can be the origin of a person's dependence, the most common being aging, accidents or chronic illnesses, and they can give rise to different types of dependence, such as physical, mental or psychic, sensory or mixed dependence.”
“All these factors can lead to a decompensation with the environment and cause situations of dependency in people with mental illness, especially in those with severe mental illness.”
line. 42 The manuscript goes so far as to state that "many people with mental illness are a part of this group". Given this, what part of the group of dependent people are you referring to? And, if only some people with mental problems, which ones?
Response:
We have rewritten the paragraphs so that it is better understood that because of the deficits they may have, they may become part of those who may be dependent on assistance with activities of daily living, especially those with severe mental illness.
Change:
“According to the International Classification of Functioning, Disability and Health (ICF), dependency is defined as the result of a deficit in bodily function (i.e., physiological functions of the body systems, including psychological functions), bringing with it a limitation in activity”
“There are several causes that can be the origin of a person's dependence, the most common being aging, accidents or chronic illnesses, and they can give rise to different types of dependence, such as physical, mental or psychic, sensory or mixed dependence.”
“All these factors can lead to a decompensation with the environment and cause situations of dependency in people with mental illness, especially in those with severe mental illness.”
line. 57 The manuscript defines a professional caregiver as the person who attends to demands for care and people in situations of dependency. I consider this definition limited, the professional caregiver has many areas, attends to needs that may not be expressed in demands and to people who may not have dependency. In any case, professional care is not linked to the Nursing profession, which is the discipline responsible for the study of care. Authors are requested to review the definition and justify this limitation.
Response:
Thank you very much. We have rewritten the definition and added several sentences so that it does not give such a simplistic concept of caregivers.
Change:
“A professional caregiver is the person who, from different fields, is in charge of carrying out functions that exceed other person's own abilities to take care of him/herself and to promote a better quality of life [5]. Their functions are usually related to the promotion of autonomy, mobility, hygiene, feeding, elimination (i.e., deposition) or safety, among others. In addition, they provide a point of support through functional social interaction [6]. He or she is generally trained in social health and…”
- Aim
line.114 The main aim includes a methodological aspect. The aim is "to identify the variables...". The qualitative analysis and systematic review should not be included in this section.
Response:
Thank you for your comment. We have removed the qualitative analysis from the objective and rewritten it, however, it is our understanding that in systematic reviews, the overall objective should indicate that, in order to be achieved, a systematic review of the literature is necessary.
Change:
“Therefore, the main aim of the current study is to identify variables related to the professional care of people with mental illness (i.e., protective or stressor variables) through a systematic review.”
-Method
2.1 Information sources
The study process with the phases carried out is not an information resource. This information would be more correct in section 2.3.
Response:
Thanks, we have moved that sentence to the section indicated by you.
Change:
“2.3. Search Strategy. The bibliographic search was carried out in three phases: an initial search to obtain an overview of the current situation, the application of inclusion and exclusion criteria, and a manual search to evaluate the results obtained. The combinations of terms used…”
- Discussion
line. 281 Clarify the concept of "mixed methods studies" because there is a type of mixed methodology that is not applied in the study. The idea can be "a set of studies with different methodologies" or something similar that the authors consider.
Response:
Thank you, we have modified this concept to the idea you have proposed.
Change:
“The aim of the study is to carry out a qualitative analysis of a set of studies with different methodologies that include empirical research or interventions to identify variables affecting the professional care exercise of people with mental illness (i.e., protective or stressor variables).”
line. 352 The manuscript states that only one researcher performs the review. This is an important limitation in the methodology. In addition, in the contribution of authors it seems that there are many people who participate in each phase of the study. Please, the authors should clarify who performs which phase.
Response:
This limitation has been overcome thanks to the Covidence program, in which another researcher accepted and rejected the different articles independently of the first researcher. In addition, we have clarified who performs each phase.
Change:
“This process was carried out by one of the authors and corroborated by another through the Covidence tool [21].”
line. 360 The manuscript states that it collects articles from "any language" but it seems that only English and Spanish are included. The authors must be rigorous in the analysis because studies in German, Chinese, Portuguese, or other languages are not included.
Response:
The search was not limited to articles written in Spanish or English, so if any article in another language had appeared, we would have included it. Although this is a complex task that requires constant contact with the translator, it is worthwhile to have a more holistic view and not limit it to one or two languages. However, none have met the inclusion and exclusion criteria.
Change:
None.
-Conclusions
The section repeats a lot of what was stated in the discussion. The authors must review it and guide it to a conclusion related to the aim, the academic and clinical impact of the research, and even future lines.
Response:
Thank you, we have rewritten part of the conclusion so that it is not so redundant and to include the aspects you have recommended. We believe it is now more complete.
Change:
“This study identifies different variables related to professional care for people with mental illness. However, it is not focused exclusively on those centered on a pathologizing view of caregiving, but also on variables that promote the well-being of the caregiver and, as a consequence, of the person being cared for, and that can act as protective factors. In this way, the aim is to raise awareness among researchers of the importance of changing the point of view and the focus of research towards a more positive and strengths-based approach of the caregiving profession. On a practical level, it is worth stressing the importance of carrying out training programs and specific training of caregivers towards mental illness, since several studies have highlighted the lack of knowledge and stigma that still prevails among these professionals towards this condition.
Research and clinical practices related to caregivers, both formal and informal, are of vital importance in all societies today since it is predicted that mental illness may become the leading cause of disability worldwide by 2030 [4]. But also because of the rise of other factors such as the aging of the population, which is expected to double in the next 30 years to 1.5 billion people by 2050 [65]. It is therefore vitally important to care for those who care, because only in this way will it be possible to provide quality care for each and every person who requires it.”
I hope that these considerations can be of help in revising the manuscript and resubmitting it. Thank you very much.
Response:
Of course, thanks to your comments the manuscript has a stronger foundation and can be understood with greater clarity. Thank you again.
Change:
None.
Round 2
Reviewer 1 Report
I appreciate all efforts that the authors made in response to my comments. The revised manuscript clarifies all the points that I raised and will help readers to understand the current manuscript.
This manuscript is a resubmission of an earlier submission. The following is a list of the peer review reports and author responses from that submission.